# Effect of Hemp Seed Oil on Milk Performance, Blood Parameters, Milk Fatty Acid Profile, and Rumen Microbial Population in Milk-Producing Buffalo: Preliminary Study

**DOI:** 10.3390/ani15040514

**Published:** 2025-02-11

**Authors:** Qichao Gu, Bo Lin, Dan Wan, Zhiwei Kong, Qinfeng Tang, Qi Yan, Xinghua Cai, Hao Ding, Guangsheng Qin, Caixia Zou

**Affiliations:** 1College of Animal Science and Technology, Guangxi University, Nanning 530004, China; guqchao@gmail.com (Q.G.); linbo@gxu.edu.cn (B.L.); wan89dan@163.com (D.W.); kongzhiwei2013@126.com (Z.K.); 899qinghui@163.com (Q.T.); yanqi7798@163.com (Q.Y.); 13367811557@163.com (X.C.); haodingxj@163.com (H.D.); 2Guangxi Key Laboratory of Animal Breeding, Disease Control and Prevention, Nanning 530004, China; 3Buffalo Research Institute Chinese Academy of Agricultural Sciences and Guangxi Zhuang Nationality Autonomous Region, Nanning 530001, China

**Keywords:** milk-producing buffalo, oil supplementation, natural additive, antioxidants, milk fatty acid profiles, rumen microbiota

## Abstract

Vegetable oils containing unsaturated fatty acids play an important role in improving milk quality and enhancing buffaloes’ overall health. Hemp seed oil (HSO), a native vegetable oil from Bama in Guangxi, China, which is known as “the hometown of longevity”, is particularly rich in unsaturated fatty acid (e.g., linoleic acid). In this study, our results show improvements in the serum antioxidant capacity and lipid metabolism regulation following dietary supplementation with HSO. Specifically, we found that HSO supplementation increased the proportion of polyunsaturated fatty acids (e.g., omega-6 and omega-3), as well as milk C18 saturated (C18:0) and unsaturated fatty acids (e.g., C18:2n6c, C18:3n3, and C18:2n9c) in milk samples. This study provides a basis for using HSO as a natural feed additive to improve the milk quality and health of milk-producing buffalo.

## 1. Introduction

In modern society, people are becoming increasingly conscious of eating a healthy diet. Hemp seed oil is a lipid vegetable oil extracted from the dried and mature seeds of hemp (*Cannabis sativa* L.). Hemp is a specialty of Bama (Guangxi, China), which is also known as the “hometown of longevity”, with a cultivation area of more than 2300 hectares, and a hemp seed yield of about 1000–2000 kg/hectare [1,2]. Notably, this oil is characterized by being the only natural vegetable oil that can be dissolved in water and has an extremely high unsaturated fatty acid (UFA) content, among which more than 80% of the total fatty acid composition is polyunsaturated fatty acids (PUFAs); additionally, over 50% of the total fatty acid composition is 18:2n-6 (linoleic acid, LA), and ~20% is 18:3n-3 (linolenic acid, ALA) [3]. It can be an important source of omega-6 and omega-3 PUFAs, which are present in an optimal nutrient ratio of 3:1 (LA to ALA) [3]. Other lipid vegetable oils (e.g., palm oil, adlay seed oil, soybean oil, coconut oil, and blended seed oil) are also rich in UFAs. When added to animal diets, these oils can improve the antioxidant capacity of animal serum by increasing antioxidant enzyme activities (the total antioxidant capacity increased by 0.57–6.56 U/mL, while the malondialdehyde and catalase contents decreased by 1.92–23.46 nmol/mL and 1.50–4.97 U/mL, respectively) [4,5]. Additionally, these oils can regulate lipid metabolism by increasing the high-density lipoprotein cholesterol content (increased by ~22.5 mg/dL) and reducing the total cholesterol and triglyceride levels in the blood (decreased by ~5 mg/dL) [6]. Hemp seed oil not only has higher levels of antioxidant properties than those of many other seed oils (e.g., linseed oil, olive oil, and niger seed oil) [7], but it has also demonstrated the ability to lower the total cholesterol (decreased by 1.28 mmol/mL) and triglyceride (decreased by 2.65 mmol/mL) levels in rats [8].

The previous studies have shown that adding sunflower seed oil (130 g/d), linseed oil (130 g/d), and babassu oil (90 g/d) to diets can improve feed intake, with the dry matter intake decreasing by 0.15 kg/d, 0.18 kg/d, and 0.46 kg/d respectively [9,10]. Moreover, linseed oil supplementation increases the yields of milk fat and milk lactose, with increases of 15 g/d and 13 g/d, respectively [9]. In addition, sunflower seed oil or linseed oil (130 g/d) supplementation can effectively regulate the milk fatty acid (FA) composition of ruminants, including a decrease in saturated fatty acids in milk, with those in milk 10:0, milk 11:0, milk 12:0, milk 13:0, milk 14:0, milk 15:0, and milk 16:0 decreasing by 3.51 g/100 g FA or 2.60 g/100 g FA, 0.08 g/100 g FA or 0.06 g/100 g FA, 2.50 g/100 g FA or 2.12 g/100 g FA, 0.15 g/100 g FA or 0.13 g/100 g FA, 4.14 g/100 g FA or 3.92 g/100 g FA, 0.50 g/100 g FA or 0.44 g/100 g FA, and 10.38 g/100 g FA or 10.72 g/100 g FA, respectively [9]. Meanwhile, regular sunflower oil (48 g/d) [11], linseed oil (48 g/d or 66 g/d) [11,12], and echium oil (~40 g/d) [13] supplementation increases the amount of functional unsaturated fatty acids in milk, such as conjugated linoleic acid (CLA), with increases of 1.22 g/100 g FA, 1.04 g/100 g FA, 1.64 g/100 g FA, and 15.51 g/100 g FA, respectively. Cozma et al., who fed dairy goats diets supplemented with 93 g/d hemp seed oil, also reported an increase in the CLA content in goat milk, with an increase of 0.18 g/100 g FA [14]. Moreover, Delavaud et al. found that both cows and goats fed diets containing hydrogenated palm oil (3.0% of total DMI; 639.3 g/d for cows and 79.5 g/d for goats) showed an increase in hydrogenated palm oil content in milk C18:0, with increases of 2.70 g/100 g FA and 3.10 g/100 g FA in the cows and the goats, respectively [15]. Rumen microbiota hydrogenate unsaturated fatty acids into saturated fatty acids through biohydrogenation, which is an important process that affects the composition of milk fatty acids [16,17]. This is an important source of saturated fatty acids in milk fat, and some biohydrogenation intermediates produced during the hydrogenation process are the source of unsaturated fatty acids in milk fat [18,19,20]. Several recent studies indicated that the addition of vegetable oil (such as olive oil, sunflower oil, coconut oil, and palm oil) affected the composition of ruminal bacteria [21,22,23]. It is unclear whether hemp seed oil, another lipid vegetable oil, can also induce changes in the bacterial composition of the dairy buffalo rumens, thereby affecting bacterial milk fatty acid metabolism.

Therefore, the aim of this study was to preliminarily evaluate the effects of dietary HSO from Bama, known as the world’s “hometown of longevity”, on milk-producing buffaloes, hypothesizing that dietary HSO could improve the milk performance, alter the blood antioxidant properties and lipid levels, modify the rumen bacterial structure, and increase the proportion of functional milk fatty acids (such as milk C18 saturated and unsaturated fatty acids) in the milk of milk-producing buffaloes.

## 2. Materials and Methods

### 2.1. Experimental Material

HSO was self-squeezed from Bama hemp seeds purchased in Bama, Guangxi, China. The hemp seeds (150 kg) were first cleaned and dried, after which they were crushed and shelled to obtain hemp kernels. These kernels (about 38 kg) were then pressed at room temperature using a press (Weiruima 120, Wuhan Weiruima Machinery Manufacturing Co., Ltd., Wuhan, China) without steaming or frying to obtain HSO. The fatty acid composition of HSO was determined according to the method described by Montserrat-de et al., with some modifications [3]. Briefly, 0.2 g of HSO was placed into a 15 mL stoppered test tube, followed by the addition of 0.5 mL of 2 mg/mL C19:0 as an internal standard and 4 mL of n-hexane. The mixture was gently shaken to dissolve the oil. Then, 5 mL of 0.5 mol/L of sodium methoxide solution was added, and the mixture was gently shaken and allowed to stand at room temperature for 10 min. A small amount of distilled water was added to the mixture, which was then gently shaken until an emulsion formed. This mixture was allowed to stand and clarify, allowing all the n-hexane solution to rise to the top. The supernatant was transferred to a dry test tube, and a small amount of anhydrous sodium sulfate was added to dry the tube. The supernatant was subsequently taken and analyzed in terms of fatty acid composition using an Agilent-7890A gas chromatograph equipped with a flame ionization detector (FID) and an HP-INNOWAX 19091N-133 capillary column (30.0 m × 0.25 mm × 0.25 μm). The fatty acid composition of HSO is shown in Table 1.

### 2.2. Experimental Design

A randomized block design was used in this research. The total experimental period lasted 42 days, including a 14-day adaptation period, followed by a 28-day formal experiment period. Seventeen healthy, four-year-old crossbred milk-producing buffalo, all of which exhibited the same parity (3) and had similar weights (BW = 580 ± 25 kg), number of days producing milk (DIM, 153 ± 10 d), and milk yields (8.56 ± 0.89 kg/d), were used in this study. The milk-producing buffalo were housed in separate fenced pens (9 m length × 4 m width) at the Buffalo Research Institute of the Chinese Academy of Agricultural Sciences in Nanning, located within China’s Guangxi Province. The milk-producing buffaloes were then divided into three dietary treatment groups (*n* = 6, 5, and 6, respectively): (1) with the addition of 0 g/d HSO (H0, *n* = 6), (2) with the addition of 100 g/d HSO (H1, *n* = 5), and (3) with the addition of 200 g/d HSO (H2, *n* = 6). HSO was added to concentrate, mixed thoroughly, and then fed individually to each milk-producing buffalo at 6:00 a.m. in the HSO treatment groups. The diets all had the same protein and energy contents. The diets, which were formulated according to the recommendations of Borghese et al. [24] (suitable for a milk yield of 8 kg/d), were provided to the milk-producing buffalo in equal amounts twice daily at 6:00 a.m. and 2:30 p.m. To facilitate the feeding of the buffaloes, balance nutrition, and improve feed utilization, the forage (including elephant grass, distilled brewer’s grain, and cassava pulp) and concentrates of the diets (forage/concentrate = 50:50) were provided separately. Each milk-producing buffalo was fed individually and had free access to water at all times. The ingredients and chemical composition of the basic diet are shown in Table 2. The samples of feed from the buffalo’s basic diet were analyzed for dry matter (method 935.29), crude protein (method 990.03), ash (method 942.05), and ether extract (method 920.39) contents [25]. The crude fiber, neutral detergent fiber, and acid detergent fiber contents were determined using an ANKOM A2000i fiber analyzer (ANKOM Technology, NY, USA). The maximum ingestion capacity of each buffalo was estimated by ensuring 10% feed refusal of a standard milk-production diet. During the last seven days of the experiment, the remaining feed in the trough was removed 30 min before daily feeding. The leftover feed was weighed, and this value was subtracted from the total weight of each feed ingredient allocated the previous day to determine the daily feed intake (subsequently converted to DMI) for each milk-producing buffalo. The DMI data are shown in Appendix A.

### 2.3. Milk Sampling and Analysis

Milk samples were collected throughout the 28–42 days of the total experiment; the milk yield is shown in Appendix A. Samples (100 mL) were collected from each experimental milk-producing buffalo at 8:30 a.m. and 3:30 p.m. The two samples from each animal were mixed thoroughly to form a mixed milk sample of 200 mL. The mixed milk sample was divided into two parts. One part (100 mL) was immediately tested for milk composition (e.g., milk fat, milk protein, total solid, solid non-fat, and lactose contents). For the different types of milk, energy-corrected milk (ECM) = milk (kg/d) + [(fat (kg/d)−40) + (protein (kg/d)−31)] × 0.01155 [26,27] and 4% fat-corrected milk (4% FCM) = 0.4 × milk (kg/d) + 15 × fat (kg/d) [26]. The milk composition and feed efficiency (milk yield/DMI; ECM/DMI) are shown in Appendix A; the other part (100 mL) was stored in a refrigerator at −80 °C to determine the fatty acid composition of milk.

Fatty acids in the milk samples were extracted for methyl esterification and gas phase detection. Based on the fatty acid determination method established by Marín et al. [13], the column selection and operation steps were followed with some modifications. An Agilent-7890A gas chromatograph was used to determine fatty acids, with a special capillary column for fatty acid methyl ester determination (Agilent HP88 (100 m × 0.25 mm × 0.20 μm)), nitrogen as the carrier gas, and a flow rate of 1.1 mL/min. The temperature of both the injection port and the detector was 250 °C. The temperature of the storage box was programmed as follows: the initial temperature was held at 150 °C for 5 min, and then increased to 175 °C at a rate of 2 °C/min and held for 15 min, increased to 200 °C at a rate of 7 °C/min and held for 20 min, and finally increased to 220 °C at a rate of 5 °C/min and held for 25 min. The sample injection volume was 1 µL, including the internal standard, and the split ratio was 20:1. The detector parameters were a hydrogen flow of 30 mL/min and an air flow 400 of mL/min.

### 2.4. Blood Sampling and Analysis

Before morning feeding on day 41 of the total experimental period, blood samples (10 mL) were collected from the carotid arteries of the milk-producing buffalo by a veterinarian. Sodium heparin tubes were chosen to collect the blood samples. Blood with sodium heparin was centrifuged at 1500× *g* for 20 min at 4 °C, and the obtained serum was then immediately stored at −20 °C for further analysis. Analysis included the measurement of total cholesterol, triglycerides, high-density lipoproteins, and low-density lipoproteins in the serum using an automatic biochemical analyzer (Hitachi 7020, Hitachi Co., Tokyo, Japan). The levels of glutathione peroxidase (GSH-Px, cat. no. A005-1), superoxide dismutase (SOD, cat. no. A001-3), catalase (CAT, cat. no. A007-1), peroxidase (POD, cat. no. A048-2), and malondialdehyde (MDA, cat. no. A003-1) and the total antioxidant capacity (T-AOC, cat. no. A015-1) in the serum were assessed using commercial assay kits (Nanjing Jiancheng Bioengineering Institute, Nanjing, China) according to the manufacturer’s instructions.

### 2.5. Rumen Fluid Sampling and Analysis

Rumen fluids were collected from the experimental milk-producing buffaloes on the last day of the experimental period before 8:30 a.m. feeding and after blood sampling. The rumen fluid was collected through a stomach tube as described by Shen et al. [28]; however, the process was performed with some modifications. The specific process is detailed in our previous study [29]. The collected rumen fluid (500 mL) was divided into two parts; one part (400 mL) was used to determine ruminal pH and NH_3_-N, lactic acid, acetate, propionate, butyrate, and total volatile fatty acid contents (these data are shown in Appendix A, while the other part (100 mL) was aliquoted into 15 mL polypropylene tubes and frozen at −80 °C for DNA extraction later.

A pH meter (PP-50-PU, Sartorius) was used to measure the ruminal pH; the colorimetric method was used to determine the NH3-N concentration [30]; and a gas chromatograph (Agilent-7890A) equipped with a flame ionization detector and a capillary column (HP-INNOWAX 19091N-133 [30.0 m × 0.25 mm × 0.25 μm]) was used to determine the lactic acid and volatile fatty acid (VFA) concentrations, including acetate, propionate, and butyrate. Briefly, during sample injection, the injection volume was 1 µL, including the internal standard, with nitrogen used as the carrier gas, and the temperature was increased from 60 °C to 200 °C at a rate of 20 °C/min, and then held at 200 °C for 3 min. The inlet and detector temperatures were set at 250 °C and 300 °C, respectively, with a split ratio of 10:1. The flow rate of hydrogen carrier gas was 25 mL/min.

### 2.6. Rumen Bacterial DNA Extraction and Sequencing

Microbial community DNA was extracted from freeze-dried rumen fluid using previously described methods [31]. The V3–V4 hypervariable region of the bacterial 16S rRNA genes were targeted using the primers 515F (5′-GTTTCGGTGCCAGCMGCCGCGGTAA-3′) and 806R (5′-GCCAATGGACTACHVGGGTWTCTAAT-3′). The 16S rRNA genes were amplified following the protocols described by Magoč and Salzberg [32]. For amplification datasets, paired-end sequencing was conducted using an Illumina MiSeq platform (Shenzhen Huada Gene Technology Co., Ltd., Shenzhen, China). The high-throughput sequence read quality and Good’s coverage estimates are shown in Appendix A. The raw sequences generated in this study are available from NCBI under the BioProject accession number PRJNA609573 for bacterial (n = 17) community datasets.

The QIIME (v1.80) pipeline was used to filter the raw sequence data as described by Anderson and Walsh [33]. UCLUST was used to conduct operational taxonomic unit (OTU) clustering at the 97% nucleotide similarity level. Then, the OTUs were assigned taxonomic classifications using the ribosomal database project classifier tool by comparing the bacterial 16S rRNA gene sequences against the Greengenes database. The OTUs were annotated at the phylum, class, order, family, and genus taxonomic levels. Five metrics were used to calculate community alpha diversity, including Chao1 and ACE species richness estimates, the count of unique OTUs (i.e., observed species), and the Simpson and Shannon diversity indices.

### 2.7. Statistical Analyses

The data for milk fatty acid profiles, blood indicators, and the relative abundances of bacteria were analyzed using SPSS (V19.0; SPSS Inc., Chicago, IL, USA). The statistical model used to analyze the data was as follows, Y =es),_i_ + T_j_ + P_k_ + ε_ijk_, where Y is the dependent variable; L is the overall mean; A_i_ is the effect of the weights, DIMs, and milk yields of the milk-producing buffaloes; T_j_ is the effect of the diet; P_k_ is the effect of the sampling period; and ε_ijk_ is the residual error term. The model included HSO supplementation as fixed effects, with the weights, DIMs, and milk yields of the milk-producing buffalo considered as a random variable. Considering that the experimental animal groups were unbalanced (6, 5, and 6), statistical differences between the data of dietary treatments were determined using a non-parametric Kruskal–Wallis test. A power test was conducted for each variable to avoid the influence of a low number of experimental units and treatments, which could increase the risk of Type I Error. *p* ≤ 0.05 was considered statistically significant, and *p* < 0.10 was considered a trend. All the results are shown as means. SEM represents the standard error of the mean.

## 3. Results

### 3.1. DMI and Milk Performance

The effect of dietary HSO on DMI is shown in Appendix A. Compared to the group without HSO supplementation, the DMI tended to decrease with dietary HSO supplementation (*p* = 0.06), with a decrease of 1.44 kg/d in both the H1 and H2 groups, respectively. Meanwhile, the milk production indicators (milk yield, energy-corrected milk (ECM), 4% fat-corrected milk, total milk solids, and non-fat milk solids), as well as the milk content parameters (milk protein, milk fat, total milk solids, non-fat milk solids, and milk lactose) and feed efficiency measures (milk yield/DMI and ECM/DMI) were not changed by dietary HSO supplementation (*p* > 0.05).

### 3.2. Antioxidant Capacity

As is shown in Table 3, compared to those of the group without HSO supplementation, the T-AOC (*p* = 0.05), CAT (*p* <0.01), and GSH-Px (*p* = 0.02) contents were significantly increased in the blood serum with dietary HSO supplementation. Specifically, the T-AOC, CAT, and GSH-Px contents increased by 0.53 U/mL and 1.16 U/mL, 0.57 U/mL and 1.15 U/mL, and 105.33 U/mL and 134.51 U/mL in the H1 and H2 groups, respectively. Meanwhile, the MDA levels significantly decreased with dietary HSO supplementation, showing decreases of 0.72 nmol/mL and 0.24 nmol/mL in the H1 and H2 groups, respectively. In addition, the POD and SOD were not affected by dietary HSO supplementation (*p* > 0.05).

### 3.3. Blood Lipid Metabolites

The effect of dietary HSO on the blood lipid metabolites in the milk-producing buffaloes is shown in Table 4. Compared to that of the group without HSO supplementation, the HDL-C content tended to increase in blood (*p* = 0.09) with dietary HSO supplementation, and it increased by 0.22 mmol/L and 0.37 mmol/L in the H1 and H2 groups, respectively. However, the total cholesterol, triglyceride, and LDL-C contents were similar among the groups (*p* > 0.05).

### 3.4. Milk Fatty Acid Composition

As shown in Table 5, compared to the group without HSO supplementation, the proportions of C18:0 (*p* = 0.02), C18:1n9t (*p* = 0.02), C18:2n6c (*p* = 0.02), C18:3n3 (*p* < 0.01), C18:2n9c (*p* = 0.04), and C20:4n6 (*p* = 0.05) were significantly increased with dietary HSO supplementation. Moreover, the proportions of C18:0, C18:1n9t, C18:2n6c, C18:3n3, C18:2n9c, and C20:4n6 increased by 2.69 g/100 g FA and 5.29 g/100 g FA, 0.83 g/100 g FA and 1.81 g/100 g FA, 0.21 g/100 g FA and 0.55 g/100 g FA, 0.03 g/100 g FA and 0.14 g/100 g FA, 0.54 g/100 g FA and 0.75 g/100 g FA, and 0.04 g/100 g FA and 0.01 g/100 g FA in the H1 and H2 groups, respectively. Similarly, the proportions of C18:1n11t (*p* = 0.07) and total CLA (*p* = 0.06) tended to increase with the dietary HSO supplementation, and the proportions of C18:1n11t and total CLA increased by 7.03 g/100 g FA and 4.73 g/100 g FA and by 0.26 g/100 g FA and 0.37 g/100 g FA in the H1 and H2 groups, respectively. Meanwhile, the proportion of omega-3 (*p* = 0.02), omega-6 (*p* = 0.02), and MUFA (*p* = 0.02) were significantly increased with dietary HSO supplementation. Specifically, the proportions of omega-3, omega-6, and MUFA increased by 0.04 g/100 g FA and 0.17 g/100 g FA, 0.25 g/100 g FA and 0.56 g/100 g FA, and 0.81 g/100 g FA and 1.48 g/100 g FA in the H1 and H2 groups, respectively. However, the proportion of C20:5n3 was significantly increased in the H2 group (*p* = 0.04), with an increase of 0.01 g/100 g FA. The omega-3/omega-6 ratio showed an upward tendency in the H2 group (*p* = 0.06), increasing to 0.21.

### 3.5. Bacterial Diversity

Five indices were used to measure the alpha diversity of the rumen bacteria, including community richness (observed species, Chao1, and ACE) and community diversity (Shannon and Simpson). It can be seen from Table 6 that among the groups, the observed species, the Chao1 estimator, the ACE estimator, the Shannon index, and the Simpson index values do not significantly differ (*p* > 0.05).

### 3.6. Bacterial Composition

The relative abundances of bacteria are significant, and two milk-producing buffalo samples were chosen for further analysis, as shown in Table 7 and Figure 1a,b. The three most dominant bacterial phyla were Bacteroidetes, Firmicutes, and Proteobacteria across all the experimental groups, and the three most dominant bacterial genera were *Prevotella*, *Acinetobacter*, and *Butyrivibrio*. Meanwhile, the relative abundance of the genus *Paludibacter* (*p* = 0.07) tended to decrease with HSO supplementation compared to that of the group without HSO supplementation, and it decreased by 0.34% and 0.35% in the H1 and H2 groups, respectively. In contrast, with dietary HSO supplementation, the relative abundances of *Acetobacter* was significantly decreased in the H1 group, while it significantly increased in the H2 group (*p* = 0.03). Specifically, the relative abundance of *Acetobacter* decreased by 0.55% in the H1 group, while it increased by 0.73% in the H2 group.

## 4. Discussion

### 4.1. DMI and Milk Performance

In this study, the DMI decreased with HSO supplementation. A decrease in DMI also occurred in ruminants that were fed diets supplemented with linseed oil [34], camelina oil [35], corn oil [36], rapeseed oil [37,38], or sunflower oil [39]. Our results are likely influenced by the following factors: (1) the FAs in HSO may have a direct inhibitory effect on autonomic feeding by inhibiting gastric smooth muscle contraction, thus inhibiting continuous rumen motility [40]; (2) HSO is rich in unsaturated FAs, which may cause disturbances in rumen function by damaging the cell membrane structure of microbial communities, leading to toxicity [41]; and (3) the energy in feed increases after adding HSO, triggering the animal’s instinct to stop eating once it has obtained sufficient energy [42]. Perhaps as a result of the differences in the FA composition and the doses of vegetable oil in diet composition or in animal species, Baluchi lambs fed diets supplemented with grapeseed oil (2% grapeseed oil in a hay-based diet) increased their DMI, but not significantly [43]. This finding is consistent with the results of this study. Barletta et al. [44] also reported a decline in milk yield with soybean oil supplementation, though not a significant one. This could be due to fat supplementation decreasing de novo synthesis in the mammary gland [45]. Another reason may be that vegetable oil supplementation affects the rumen microbial structure, which, in turn, leads to changes in rumen fermentation [46]. However, in this study, no significant changes in milk composition, feed efficiency, or fermentation parameters (Appendix A) were found with HSO supplementation. Therefore, dietary HSO from the “longevity village” of Bama did not negatively affect the rumen fermentation or the performance of milk-producing buffaloes, but like other vegetable oils, it did decrease the DMI in the milk-producing buffaloes.

### 4.2. Antioxidant Capacity

Animals have an endogenous antioxidant protection system that can help scavenge reactive oxygen species (ROS) generated by lipid oxidative stress to prevent damage to cells and tissues. T-AOC is the cumulative effect of all antioxidants in sera. In the present study, dietary HSO increased the T-AOC content, and this was the highest in the H2 group. Signor et al. also reported a similar increase in the T-AOC content in sera of rats fed a diet supplemented with adlay seed oil [47]. Furthermore, in this study, the CAT and GSH-Px contents also increased with HSO addition, and it was the highest in the H2 group. GSH-Px and CAT have good synergistic antioxidant effects, and both can catalyze the decomposition of H_2_O_2_ into H_2_O and O_2_, thereby scavenging H_2_O_2_ [48,49]. Additionally, MDA acts as a marker of the degree of lipid peroxidation in the body and indirectly reflects the extent of cellular damage [50]. In this study, the MDA content increased with HSO addition and was the lowest in the H1 group. Similarly, rats fed a diet with adlay seed oil (1.25 mL/kg and 5 mL/kg) showed decreased MDA levels in the sera [4], while chickens at 21 days of age fed a diet with palm oil also exhibited a quadratic reduction in the serum MDA content, with the lowest observed in 2% palm oil treatments [5]. These results may be attributed to the following factors: (1) The addition of HSO increases the PUFA content, which, in turn, can lead to the increased production of free radicals, thereby increasing the possibility of spontaneous lipid oxidation [51]. For this reason, when animals are fed a diet high in PUFAs, the body induces an increase in the activity of endogenous antioxidant enzymes to cope with the resulting oxidative stress [52]. (2) The CLA in HSO is considered an antioxidant that inhibits the formation of peroxides from PUFAs [53]. (3) The HSO used in this study was obtained by cold-pressing and may contain a large amount of antioxidant compounds, such as phytosterols and lower-molecular-weight phenolic acids [54]. This indicated that similar to other vegetable oils, dietary HSO from Bama, known as the “longevity village”, could notably improve the antioxidant capacity of milk-producing buffaloes, with some indicators of antioxidant activity being affected by its dosage.

### 4.3. Blood Lipid Metabolites

In this study, dietary HSO increased the HDL-C content in the blood of the milk-producing buffaloes. Chen et al. also reported an increase in HDL-C levels in the sera of mice fed a high-fat diet supplemented with blended oils (including Zanthoxylum bungeanum seed oil, walnut oil, camellia seed oil, and perilla seed oil) [55], with similar results observed in mice fed a diet supplemented with coconut oil [6]. HDL-C plays an important role in transporting cholesterol from extrahepatic tissues to the liver for metabolism, thereby preventing various diseases caused by excess cholesterol [56]. The observed increase in HDL-C levels in the blood may be attributed to the fact that HSO increases the proportion of saturated fatty acids in the diet, especially myristic acid [57]. Elevated levels of HDL-C in the blood can help prevent the occurrence of fatty liver in milk-producing buffaloes supplemented with HSO from the village of Bama.

### 4.4. Milk Fatty Acid Composition

Other studies have confirmed that the fatty acid profile of milk can be extensively modified by supplementing diets with vegetable oils [58,59]. In particular, the content of some long-chain fatty acids (≥C18:0) in milk increased with the addition of vegetable oils to diets [34,60]. In the present study, the proportions of long-chain fatty acids (e.g., C18:0, C18:1n9t, C18:2n6c, C18:3n3, C18:2n9c, and C20:4n6) increased with dietary HSO supplementation. Meanwhile, the content of other long-chain fatty acids, such as C18:1n11t, showed an upward tendency. Similarly, supplementation with many other vegetable oils, such as soybean oil (Gómez-Cortés et al., who studied goats fed alfalfa hay-based diets supplemented with 6% soybean oil) [61], sunflower oil (Marín et al., who studied lactating dairy goats fed diets supplemented with 48 g/d, and Ollier et al., who studied dairy goats fed low-forage diets supplemented with 130 g/d) [11,62], and linseed oil (Marín et al., who studied lactating dairy goats fed diets supplemented with 48 g/d or 66 g/d), increased the proportion of long-chain (≥C18:0) fatty acids in milk fat [12,13]. The reason for this is that HSO from the village of Bama is rich in long-chain unsaturated fatty acids, which results in increased absorption and the secretion of dietary or rumen-derived FAs [63].

In this study, dietary HSO increased the proportions of C18:3n3. Some studies by Mare et al. found a significant increase in the proportion of C18:3n3 in goat milk fat with high dosages of linseed oil (48 g/d or 66 g/d) fed to goats [12,13]. However, Assaf ewes fed diets supplemented with sunflower oil (2% DM, alfalfa hay-based diets) had decreased milk C18:3n3 levels [64]. The reasons for this may be attributed to diet composition, oil composition, the dosage, or the animal species. At the same time, the proportion of C18:2n6c in milk was also highest in the H2 group. ALA and LA are essential fatty acids that can only be obtained by eating food, and high doses of HSO are rich in these two polyunsaturated fatty acids, as shown in Table 1. In the rumen, C18:0 is a saturated fatty acid formed after the hydrogenation of LA or ALA, while the biohydrogenation intermediate 18:2n9c is only formed via the rumen hydrogenation of LA, and then absorbed through the mammary glands [20]. Therefore, the contents in milk C18:0 and C18:2n9c were the highest in the H2 group. Moreover, in this study, all of the CLA was found in milk C18:2n9c and C18:2n10t, and the content showed an increasing trend after the addition of HSO. Similarly, studies using high levels of soybean oil (3% or 5% soybean oil in the DM of diets) [65,66] and sunflower oil (5.1% sunflower oil in the DM of diets) also reported similar findings [67]. This is possibly due to the isomerization of LA and ALA to CLA in the rumen via the biohydrogenation pathway caused by HSO from the “longevity village” of Bama, which is then absorbed into milk via the mammary glands [68]. Another possibly reason is that CLA is synthesized in the mammary glands of lactating ruminants using oleic acid (C18:1) as a precursor and delta 9-desaturase enzymes [20], and the addition of HSO from Bama provides more C18:1 (Table 1). There are also studies showing that the CLA in diets is directly absorbed through the mammary glands and into milk [69].

In addition, the dietary supplementation of HSO increased the proportions of omega-3 and omega-6. Vargas-Bello-P also reported that the supplementation (3% DM) of diets with soybean oil increases the n-3 and n-6 polyunsaturated fatty acid contents [65]. This is due to the biosynthesis of omega-3 and omega-6 fatty acids through an enzymatic desaturation and elongation process using ALA and LA provided by high doses of HSO from the village of Bama [70]. Both omega-3 and omega-6 are important functional fatty acids for human health; however, they must be consumed in their proper ratio (1:4–1:1) to achieve the desired results, which include improving immunity and promoting cardiovascular health. In the present study, the omega-3/omega-6 ratio increased from 0.18 in the H0 group to 0.21 in the H2 group. A similar change was also found in goats fed diets with linseed oil [12]. This is possibly due to an increased n-3/n-6 ratio, which may also help convert alpha-linolenic acid into long-chain n-3 PUFAs (e.g., DHA), albeit with reduced competition for elongase [70,71]. At the same time, this explains why the proportion of DHA was highest in the H2 group.

Considering the composition of MUFAs, these results also explained why the proportions of MUFA increased after dietary supplementation with HSO. A similar result was also reported by Bernard et al., who found that high doses of sunflower seed oil or linseed oil (130 g/d) added to diets, whether grass hay or maize silage, increased the MUFA contents in goat milk fatty acids [9]. Alfalfa hay-based diets supplemented with high doses of high oleic sunflower oil, regular sunflower oil, or linseed oil (48 g/d) showed consistent results [12].

### 4.5. Bacterial Diversity and Composition

Consistent with our results, Tapio et al. observed that the bacterial diversity in the rumens of dairy cows did not significantly differ with sunflower oil supplementation [72]. Similar results were also reported for lambs fed diets supplemented with linseed oil [73] and Holstein male calves fed diets supplemented with olive oil, sunflower oil, coconut oil, and other vegetable oils [21,22,23]. In contrast, high doses of tucumã oil (1%, *v*/*v*) supplemented in diets using the nylon bag in situ method resulted in a decrease in the Shannon index of the ruminal bacteria [74]. This variation may be attributed to the differences in diet composition, oil composition, doses, or sampling methods.

A 16S rRNA gene relative abundance ≥ 0.10% can be considered the threshold for identifying active core rumen bacterial populations [75]. Thus, the active core bacterial community consisted of six phyla and 11 genera in this study. Similar to our study, whether or not the diets were amended with vegetable oil, Firmicutes and Bacteroidetes have been found to be the top two dominant taxa in the rumens of Holstein dairy cows [21,22,23], followed by Proteobacteria and other phyla [22,23]. Accordingly, Cancino-Padilla et al., Hu et al., and Liu et al. also observed that *Prevotella* was dominant in dairy cow rumens regardless whether the dairy cows were fed diets with or without vegetable oil supplementation [21,22,23]. Notably, the genera *Prevotella*, *Acinetobacter*, and *Butyrivibrio* belong to the Bacteroidetes, Firmicutes, and Proteobacteria phyla, respectively. The results indicate that the addition of HSO from Bama did not notably affect the dominant rumen populations at the phylum level in the milk-producing buffaloes.

A previous study documented a significant difference in the bacterial community composition in cattle and lamb rumens after their diets were supplemented with vegetable oil compared to that without vegetable oil [73,76]. However, the bacteria exhibited extensive genetic diversity within higher-level taxonomic levels (e.g., at the phylum and family levels). Thus, to understand the functions of bacteria in rumens after dietary changes, it is necessary to evaluate compositions at a finer taxonomic resolution, such as the genus level [77]. Furthermore, it is particularly important to investigate the active core rumen bacteria that may play important roles, but exhibit lower abundances [78]. Considering the less abundant taxa in this study, the relative abundances of the genus *Paludibacter* decreased with dietary HSO supplementation. This decrease was likely due to the antibacterial effects of vegetable oils from plant seeds with high concentrations of unsaturated fatty acid [46,79]. *Paludibacter* is associated with cellulolysis and hydrogenotrophic methanogens [80]. The inhibition of methane production by long-chain unsaturated fatty acids primarily alters the metabolic use of hydrogen and carbon dioxide by acting as a receptor for hydrogen, or via direct toxic effects on rumen microorganisms [16]. In addition, the relative abundances of *Acetobacter* were highest in the H2 group. Li et al. observed that cellulase activity was efficiently synthesized extracellularly by certain strains of *Acetobacter* [81]. The role of *Acetobacter* in rumen metabolism is not well understood, although their production of acetate, a precursor substance for the synthesis of milk fat, mainly occurs due to certain organic acids (e.g., lactic acid) and ethanol [82]. Notably, Khanna et al. pointed out that acetic acid bacteria (such as *Acetobacter* species) can use glycerol [83], a product of the rumen microbial hydrolysis of exogenous lipids [84], to synthesize DHA.

## 5. Conclusions

This preliminary study investigated the changes in the milk production performance, the milk fatty acid profiles, the blood parameters, rumen fermentation, and the rumen bacterial communities in milk-producing buffaloes fed diets supplemented with HSO, a native vegetable oil from the “longevity village” of Bama (Guangxi, China). These data suggest that dietary supplementation with HSO improves antioxidant capacity by increasing the T-AOC, CAT, and GSH-Px levels and decreasing the MDA levels in serum, regulate blood lipid metabolism by increasing the HDL-C content, and increases the proportion of functional fatty acids in milk, including C18 saturated and unsaturated fatty acids, omega-6, and omega-3. Meanwhile, only a few less-abundant bacteria (e.g., *Acetobacter*) showed significant changes at the genus level. Although the addition of HSO decreased the DMI, it did not affect feed efficiency, the milk production performance, or rumen fermentation. Moreover, some of the observed parameters in this study were affected by the dosage of HSO, indicating that further research is needed to determine the optimal dosage for milk-producing buffaloes.

## Figures and Tables

**Figure 1 animals-15-00514-f001:**
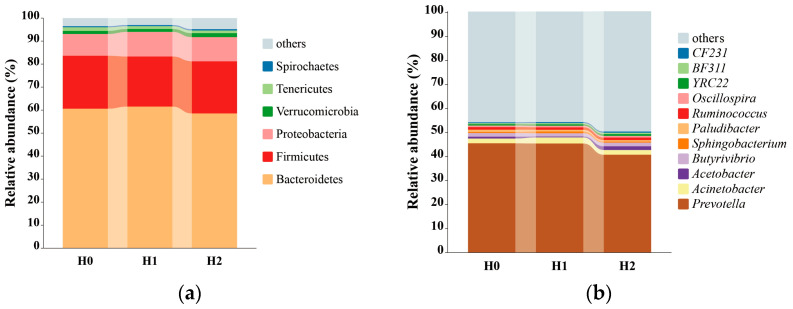
Relative abundances of dominant bacterial populations at phylum (**a**) and genus (**b**) levels after exposure to three dietary treatments. H0, without HSO supplementation group; H1, group with 100 g/d HSO supplementation; H2, group with 200 g/d HSO supplementation.

**Table 1 animals-15-00514-t001:** Composition of fatty acids in HSO (g/100 g of fatty acid).

Fatty Acid	Content
C14:0	0.012
C16:0	6.301
C16:1n7	0.094
C18:0	2.918
C18:1n9t	0.001
C18:1n11t	0.021
C18:1n9c	10.238
C18:2n6c	58.321
C18:3n6	0.476
C20:0	0.769
C18:3n3	20.243
C18:2n9c	0.365
C22:0	0.226
C20:4n6	0.015
SFA ^1^	10.226
USFA ^2^	86.774

^1^ SFA, saturated fatty acid, ^2^ USFA, unsaturated fatty acid.

**Table 2 animals-15-00514-t002:** Composition (%, as fed) and nutrients (%, DM) in basal diets.

Item	Basal Diet
Ingredients	
Maize grain	13.31
Wheat bran	3.33
Soy bean meal	4.99
Cottonseed meal	2.77
Rapeseed meal	1.66
Stone powder	0.28
CaHPO_4_·2H_2_O	0.28
NaHCO_3_	0.42
NaCl	0.42
Premix ^1^	0.28
Distilled brewer’s grain	23.90
Elephant grass	11.31
Cassava pulp	37.05
Nutritional level of diet	
Dry matter	46.60
Crude protein	17.65
Crude fiber	45.55
Acid detergent fiber	18.20
Neutral detergent fiber	37.14
Ether extract	3.34
Ash	8.01
NE_L_, MJ/kg ^2^	4.57

NE_L_ = net energy for lactation. ^1^ The composition of the premix is as follows: VA 500 KIU/kg; VD3150 KIU/kg; VE 3000 IU/kg; Fe 4.0 g/kg; Cu 1.3 g/kg; Mn 3.0 g/kg; Zn 6.0 g/kg; I 80 mg/kg; Se 50 mg/kg; and Co 80 mg/kg. ^2^ NEL was calculated according to the Feed Database in China (2022). The formula for calculating NEL is as follows: NEL (MJ/kg) = 0.1025 × TDN% − 0.502, TDN% = 1.15 × CP% + 1.75 × EE% + 0.45 × CF% + 0.0085 × NDF%2 + 0.25 × NFE% − 3.4. The other nutritional contents of the diet are measured values.

**Table 3 animals-15-00514-t003:** Effect of HSO supplementation on the serum antioxidant capacity in milk-producing buffaloes.

Item ^1^	Parameters ^2^					
	T-AOC (U/mL)	MDA (nmol/mL)	POD (U/mL)	CAT (U/mL)	SOD (U/mL)	GSH-Px (U/mL)
H0	3.06	2.36	11.11	45.14	41.4	838.81
H1	3.59	1.64	9.28	45.71	38.52	944.14
H2	4.22	2.12	10.39	46.29	44.26	973.32
SEM ^3^	0.188	0.122	0.72	0.21	1.221	18.491
*p*-value ^4^	0.05	0.02	0.72	<0.01	0.14	0.02
Power of tests						
H0 vs. H1	0.297	0.882	0.182	0.606	0.222	0.828
H0 vs. H2	0.686	0.131	0.062	0.520	0.140	0.939
H2 vs. H3	0.297	0.318	0.084	0.438	0.372	0.167

^1^ Diets supplemented with 0 g/d (H0), 100 g/d (H1), and 200 g/d (H2) of HSO. ^2^ T-AOC, total antioxidant capacity; MDA, malondialdehyde; POD, peroxidase; CAT, catalase; SOD, superoxide dismutase; GSH-Px, glutathione peroxidase. ^3^ SEM, standard error of mean. ^4^ *p* ≤ 0.05 is considered statistically significant.

**Table 4 animals-15-00514-t004:** Effect of HSO supplementation on blood lipid metabolites in milk-producing buffaloes.

	Parameters ^2^			
Item ^1^	Total Cholesterol (mmol/L)	Triglycerides (mmol/L)	HDL-C (mmol/L)	LDL-C (mmol/L)
H0	3.14	0.33	1.62	0.56
H1	2.58	0.39	1.84	0.61
H2	2.54	0.38	1.99	0.52
SEM ^3^	0.363	0.013	0.073	0.039
*p*-value ^4^	0.24	0.19	0.09	0.80
Power of tests				
H0 vs. H1	0.073	0.415	0.246	0.067
H0 vs. H2	0.078	0.475	0.551	0.095
H2 vs. H3	0.054	0.058	0.121	0.114

^1^ Diets supplemented with 0 g/d (H0), 100 g/d (H1), and 200 g/d (H2) of HSO. ^2^ HDL-C, high-density lipoprotein cholesterol; LDL-C, low-density lipoprotein cholesterol. ^3^ SEM, standard error of mean. ^4^ *p* ≤ 0.05 is considered statistically significant.

**Table 5 animals-15-00514-t005:** Effects of HSO supplementation on milk fatty acid profiles in milk-producing buffaloes.

	Treatment ^2^					Power of Tests		
Fatty Acid (g/100 g FA) ^1^	H0	H1	H2	SEM ^3^	*p*-Value ^4^	H0 vs. H1	H0 vs. H2	H2 vs. H3
C4:0	0.17	0.18	0.22	0.009	0.05	0.073	0.599	0.550
C6:0	0.71	0.79	0.75	0.034	0.57	0.234	0.065	0.067
C8:0	0.67	0.78	0.65	0.036	0.35	0.417	0.055	0.215
C10:0	1.45	1.78	1.44	0.093	0.30	0.365	0.050	0.226
C12:0	1.92	2.33	1.97	0.108	0.27	0.392	0.050	0.232
C14:0	9.18	10.69	9.46	0.440	0.16	0.319	0.055	0.180
C14:1n5	1.77	1.30	1.05	0.213	0.14	0.103	0.178	0.440
C16:0	26.74	30.71	28.27	1.010	0.28	0.345	0.081	0.167
C16:1n7	3.89	1.95	1.45	0.739	0.16	0.125	0.171	0.581
C18:0	9.85	12.54	15.14	0.814	0.02	0.368	0.746	0.338
C18:1n9t	2.35	3.18	4.16	0.269	0.02	0.261	0.834	0.555
C18:1n11t	19.80	26.83	24.55	1.296	0.07	0.699	0.274	0.118
C18:1n9c	2.30	0.85	1.44	0.472	0.04	0.153	0.085	0.910
C18:2n6c	1.89	2.10	2.44	0.088	0.02	0.440	0.627	0.356
C18:3n6	0.04	0.04	0.05	0.002	0.36	0.130	0.347	0.347
C20:0	0.18	0.12	0.05	0.049	0.06	0.066	0.131	0.222
C18:3n3	0.32	0.35	0.46	0.018	<0.01	0.218	0.989	0.919
C18:2n9c	1.03	1.57	1.78	0.135	0.04	0.462	0.711	0.091
C18:2n10t	0.25	0.24	0.24	0.023	0.93	0.052	0.052	0.050
C22:0	0.05	0.07	0.06	0.005	0.31	0.876	0.095	0.095
C20:4n6	0.13	0.17	0.14	0.008	0.05	0.797	0.073	0.364
C20:5n3	0.02	0.01	0.03	0.004	0.04	0.053	0.168	0.508
C22:6n3	0.03	0.03	0.04	0.005	0.48	0.050	0.095	0.168
Total CLA	0.64	0.90	1.01	0.070	0.06	0.404	0.661	0.088
SFA	50.93	60.00	58.01	2.111	0.08	0.483	0.199	0.068
USFA	33.82	38.62	37.84	1.462	0.24	0.260	0.142	0.055
MUFA	3.70	4.51	5.18	0.232	0.02	0.618	0.740	0.203
PUFA	30.12	34.11	32.65	1.374	0.26	0.190	0.089	0.075
SFA:USFA	1.59	1.57	1.62	0.104	0.72	0.051	0.051	0.053
Omega-3	0.36	0.40	0.53	0.023	0.01	0.377	0.918	0.803
Omega-6	2.02	2.27	2.58	0.091	0.02	0.489	0.606	0.295
Omega-3:Omega-6	0.18	0.17	0.21	0.007	0.06	0.084	0.318	0.976

^1^ CLA, conjugated linoleic acid; SFA, saturated fatty acid; USFA, unsaturated fatty acid; MUFA, monounsaturated fatty acid; PUFA, polyunsaturated fatty acid; any fatty acid(s) not detected were removed from this table. ^2^ Diets supplemented with 0 g/d (H0), 100 g/d (H1), and 200 g/d (H2) of HSO. ^3^ SEM, standard error of mean. ^4^ *p* ≤ 0.05 is considered statistically significant.

**Table 6 animals-15-00514-t006:** Statistical α-diversity values for bacterial communities in milk-producing buffalo ruminal fluids after exposure to three dietary treatments.

	Parameters ^2^				
Items ^1^	Observed Species	Shannon	Simpson	Chao1	ACE
H0	947.5	7.62	0.98	1384.13	1429.81
H1	906.8	7.49	0.98	1268.37	1334.93
H2	947	7.63	0.98	1344.55	1412.68
SEM ^3^	14.263	0.051	0.001	26.505	26.703
*p*-value ^4^	0.24	0.52	0.38	0.22	0.28
Power of tests					
H0 vs. H1	0.140	0.119	0.050	0.346	0.210
H0 vs. H2	0.050	0.051	0.050	0.082	0.056
H2 vs. H3	0.468	0.284	0.207	0.307	0.352

^1^ ACE, abundance-based coverage estimator. ^2^ Diets supplemented with 0 g/d (H0), 100 g/d (H1), and 200 g/d (H2) of HSO. ^3^ SEM, standard error of mean. ^4^ *p* ≤ 0.05 is considered statistically significant.

**Table 7 animals-15-00514-t007:** Relative abundances of dominant bacterial populations (≥0.10% abundance across all 17 milk-producing buffalo samples) at genus and phylum levels after exposure to three dietary treatments.

Taxonomy	Relative Abundance (%) ^1^					Power of Tests		
	H0	H1	H2	SEM ^2^	*p*-Value ^3^	H0 vs. H1	H0 vs. H2	H2 vs. H3
Phyla								
Bacteroidetes	60.60	61.52	58.54	1.797	0.71	0.053	0.091	0.077
Firmicutes	23.04	21.80	22.70	0.779	0.84	0.074	0.054	0.068
Proteobacteria	9.44	10.65	10.47	1.416	0.80	0.057	0.072	0.050
Verrucomicrobia	1.38	1.31	1.74	0.199	0.38	0.052	0.088	0.158
Tenericutes	1.55	1.21	1.17	0.087	0.23	0.285	0.361	0.057
Spirochaetes	0.48	0.52	0.59	0.034	0.32	0.067	0.278	0.110
Genera								
*Prevotella*	45.54	45.42	40.77	1.938	0.54	0.050	0.154	0.157
*Acinetobacter*	1.95	2.65	1.95	0.402	0.91	0.078	0.050	0.074
*Acetobacter*	0.82	0.27	1.55	0.218	0.03	0.278	0.243	0.761
*Butyrivibrio*	1.48	1.34	1.51	0.102	0.76	0.075	0.052	0.083
*Sphingobacterium*	0.46	0.83	0.47	0.143	0.95	0.116	0.050	0.102
*Paludibacter*	0.94	0.60	0.59	0.066	0.07	0.481	0.613	0.051
*Ruminococcus*	1.09	1.07	1.08	0.059	0.62	0.055	0.050	0.050
*Oscillospira*	0.61	0.63	0.67	0.033	0.99	0.055	0.103	0.069
YRC22	0.61	0.62	0.77	0.050	0.51	0.051	0.252	0.149
BF311	0.39	0.51	0.55	0.042	0.55	0.199	0.405	0.059
CF231	0.39	0.45	0.53	0.045	0.53	0.112	0.197	0.085

^1^ Diets supplemented with 0 g/d (H0), 100 g/d (H1), and 200 g/d (H2) of HSO. ^2^ SEM, standard error of mean. ^3^ *p* ≤ 0.05 is considered statistically significant.

## Data Availability

The data in this article are available upon request from the corresponding author.

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
