# Peer review of "Effect of Hemp Seed Oil on Milk Performance, Blood Parameters, Milk Fatty Acid Profile, and Rumen Microbial Population in Milk-Producing Buffalo: Preliminary Study"

_animals, 2025, doi:10.3390/ani15040514_

Round 1

Reviewer 1 Report

Comments and Suggestions for Authors

Dear authors, respectfully, it seems to me that this is a good manuscript. However, many points affect the quality of the manuscript. My main concern is about the number of experimental units. The actual number of experimental units does not support the necessary number of degrees of freedom for a correct study. It is necessary for the authors to add the power test for each variable of the study. In this way they will be able to demonstrate that the low number of experimental units and treatments did not influence the study, promoting the appearance of Type I Error. Some suggestions are below.

Simple summary: This topic should showcase the highlights of the manuscript, thereby enhancing it to make this text more eye-catching.

Lines 18-19: Irrelevant. Remove it or rewrite it.

Abstract: Improve material and methods. Add statistical design. Add real p-values. Add statistical analysis method. Improve description of results with numbers. Improve description of conclusions.

Line 26: Simple summary, abstract and body of the manuscript are different texts; in this sense, abbreviations must be fully described the first time they are used in each of the topics.

Lines 27-28: Add the breed instead of the text “dairy buffalo”.

Line 28: Please add more information about the buffaloes: breed, initial body weight, lactation period, number of calves, etc.

Lines 30-31: In this regard, were only 27 days used to evaluate the effect of the diet?

Line 35: This is not the correct description of a quadratic effect.

Line 39: What do you mean by “strongly” and “few”? Avoid informal descriptions.

Lines 42-45: This conclusion is not supported by the description of the results in the abstract.

.

Keyword: My suggestion is to eliminate all keywords and describe others. Use keywords other than the title and review the keyword suggestion in the author's instructions.

Introduction: I like this introduction because it has relevant information. My suggestion is to add numbers to improve your description because in the current form it appears a generic description. Also, avoid texts with long descriptions because these can be difficult to read. The description of HSO needs to be improved to understand the relevance of this alternative ingredient in ruminant feed (availability, production, etc.).

Lines 64-65: How much?

Line 68: How much?

Lines 68-70: Irrelevant. Remove it or rewrite it.

Lines 74-79: References should be described according to the author's instructions.

Lines 81-82: These two ideas need to be correlated. Improve the description.

Line 90: Remove it.

Material and methods

Lines 99-103: Add numbers throughout this process; numbers to understand how much is obtained in each process.

Line 117: The experimental design was not described.

Lines 118-119: In this regard, were only 27 days used to evaluate the effect of the diet?

Line 119: Add the breed of the animals.

Line 121: Were the buffaloes housed in individual stalls? What were their dimensions?

Lines 125-126: How was the oil supplied to the animals individually?

Lines 126-127: I can't find this description in the introduction.

Lines 127-128: Isoproteic and isoenergetic diets? For what amount of milk production are they formulated? Describe it here.

Lines 129-130: Why? This could influence the experiment and nutrient intake. What forage was used? What was the ratio of forage:concentrate? Describe it here.

Line 132: Use “feed” instead of “food” here and throughout the text.

Lines 137-139: How many refusals did they allow themselves?

Line 142: Please add the collection days instead of the 14 day description. In this way, it appears that the milk was collected every day for the 14 days.

Line 153: Were fatty acid standards used? Internal, external, both…

Line 180: After blood sampling?

Lines 183-186: What about the other part? How much rumen fluid was collected in total?

Lines 188-196: Were fatty acid standards used? Internal, external, both…

Line 227: The quadratic effect can be considered flawed because the study only has three treatments.

Dear authors, I have a great concern about the number of experimental units. The actual number of experimental units does not support the necessary number of degrees of freedom for a correct study. It is necessary for the authors to add the power test for each variable of the study. In this way they will be able to demonstrate that the low number of experimental units and treatments did not influence the study, promoting the appearance of Type I Error.

Results. The results description is generic. Improve your writing style showing your data in other forms more than a simple description of tables. Was higher? How much (%, g, l, etc)?

Lines 233-234: These results are not shown in the results. No P-values ​​from the regression were observed in the supplementary tables.

Lines 241-243: Why were “lin” and “quad” described in parentheses, if “linearly” and “quadratic” were described in the description? The same for all the results description.

Table 1 and 2: This table can be placed in the materials and methods topic.

For linear and quadratic effects, regression equations should be used to describe the results.

Tables are not complete; these do not show p-values ​​for comparisons of means or p-values ​​for regression. Tables should present both p-value information for comparison of means and for regression analysis.

Discussion: This discussion follows a correct description of the discussion, theories, hypotheses, or claims about how the authors obtained their results. Furthermore, the authors should decrease comparison with many authors, mainly those who have different results due to the composition of the FAs.

Lines 315-316: Ok but the milk production was not affected.

Lines 321-326: That's true, but it needs to be described in detail. Also, given the other results obtained, are you sure that these three hypotheses are valid?

Lines 335-336: This is true; in this sense, why is the above description about lower milk production necessary?

Conclusion: To improve the conclusion, add feed efficiency or feed conversion ratio to the results to show that HSO was efficient.

References: More than 50% of references are older than 10 years. Update references to reduce this value to around 20%.

Author Response

Dear authors, respectfully, it seems to me that this is a good manuscript. However, many points affect the quality of the manuscript. My main concern is about the number of experimental units. The actual number of experimental units does not support the necessary number of degrees of freedom for a correct study. It is necessary for the authors to add the power test for each variable of the study. In this way they will be able to demonstrate that the low number of experimental units and treatments did not influence the study, promoting the appearance of Type I Error. Some suggestions are below.

Answer: Thank you for your valuable suggestions. We have made the necessary revisions to the relevant sections of the manuscript.

Simple summary: This topic should showcase the highlights of the manuscript, thereby enhancing it to make this text more eye-catching.

Lines 18-19: Irrelevant. Remove it or rewrite it.

Answer: This has been revised. Please refer to lines 18–21.

Abstract: Improve material and methods. Add statistical design. Add real p-values. Add statistical analysis method. Improve description of results with numbers. Improve description of conclusions.

Line 26: Simple summary, abstract and body of the manuscript are different texts; in this sense, abbreviations must be fully described the first time they are used in each of the topics.

Answer: This has been revised. Please refer to line 26.

Lines 27-28: Add the breed instead of the text “dairy buffalo”.

Answer: This has been revised. Please refer to line 28–29.

Line 28: Please add more information about the buffaloes: breed, initial body weight, lactation period, number of calves, etc.

Answer: This has been revised. Please refer to lines 28–30.

Lines 30-31: In this regard, were only 27 days used to evaluate the effect of the diet?

Answer: This has been revised. Please refer to lines 33–34.

Line 35: This is not the correct description of a quadratic effect.

Answer: This has been revised. Please refer to lines 34–50.

Line 39: What do you mean by “strongly” and “few”? Avoid informal descriptions.

Answer: This has been revised. Please refer to lines 44–50.

Lines 42-45: This conclusion is not supported by the description of the results in the abstract.

Answer: This has been revised. Please refer to lines 50–53.

Keyword: My suggestion is to eliminate all keywords and describe others. Use keywords other than the title and review the keyword suggestion in the author's instructions.

Answer: This has been revised. Please refer to lines 54–55.

Introduction: I like this introduction because it has relevant information. My suggestion is to add numbers to improve your description because in the current form it appears a generic description. Also, avoid texts with long descriptions because these can be difficult to read. The description of HSO needs to be improved to understand the relevance of this alternative ingredient in ruminant feed (availability, production, etc.).

Answer: This has been revised. Please refer to lines 61–63.

Lines 64-65: How much?

Answer: This has been revised. Please refer to lines 70–72.

Line 68: How much

Answer: This has been revised. Please refer to lines 75–80.

Lines 68-70: Irrelevant. Remove it or rewrite it.

Answer: This has been removed as suggested.

Lines 74-79: References should be described according to the author's instructions.

Answer: This has been revised. Please refer to lines 81–100.

Lines 81-82: These two ideas need to be correlated. Improve the description.

Answer: This has been revised. Please refer to lines 101–103.

Line 90: Remove it.

Answer: This has been removed as suggested.

Material and methods

Lines 99-103: Add numbers throughout this process; numbers to understand how much is obtained in each process.

Answer: This has been revised. Please refer to lines 120–121.

Line 117: The experimental design was not described.

Answer: This has been revised. Please refer to line 141.

Lines 118-119: In this regard, were only 27 days used to evaluate the effect of the diet?

Answer: This has been revised. Please refer to lines 141–142.

Line 119: Add the breed of the animals.

Answer: This has been revised. Please refer to line 143.

Line 121: Were the buffaloes housed in individual stalls? What were their dimensions?

Answer: This has been revised. Please refer to lines 145–146.

Lines 125-126: How was the oil supplied to the animals individually?

Answer: This has been revised. Please refer to lines 150–152.

Lines 126-127: I can't find this description in the introduction.

Answer: This has been revised as suggested.

Lines 127-128: Isoproteic and isoenergetic diets? For what amount of milk production are they formulated? Describe it here.

Answer: This has been revised. Please refer to lines 152–154.

Lines 129-130: Why? This could influence the experiment and nutrient intake. What forage was used? What was the ratio of forage:concentrate? Describe it here.

Answer: This has been revised. Please refer to lines 155–157.

Line 132: Use “feed” instead of “food” here and throughout the text.

Answer: This has been revised. Please refer to line 160.

Lines 137-139: How many refusals did they allow themselves?

Answer: This has been revised. Please refer to lines 164–165.

Line 142: Please add the collection days instead of the 14 day description. In this way, it appears that the milk was collected every day for the 14 days.

Answer: This has been revised. Please refer to line 179.

Line 153: Were fatty acid standards used? Internal, external, both…

Answer: This has been revised. Please refer to line 200.

Line 180: After blood sampling?

Answer: This has been revised. Please refer to lines 217–218.

Lines 183-186: What about the other part? How much rumen fluid was collected in total?

Answer: This has been revised. Please refer to line 221 and line 223.

Lines 188-196: Were fatty acid standards used? Internal, external, both…

Answer: This has been revised. Please refer to line 231.

Line 227: The quadratic effect can be considered flawed because the study only has three treatments.

Answer: This has been revised. Please refer to line 264.

Dear authors, I have a great concern about the number of experimental units. The actual number of experimental units does not support the necessary number of degrees of freedom for a correct study. It is necessary for the authors to add the power test for each variable of the study. In this way they will be able to demonstrate that the low number of experimental units and treatments did not influence the study, promoting the appearance of Type I Error.

Answer: This has been revised. Please refer to line 267.

Results. The results description is generic. Improve your writing style showing your data in other forms more than a simple description of tables. Was higher? How much (%, g, l, etc)?

Answer: This has been revised. Please refer to the Results section of the manuscript.

Lines 233-234: These results are not shown in the results. No P-values from the regression were observed in the supplementary tables.

Answer: Thank you for your suggestions. The results are shown in the Supplementary Materials (Table S1).

Lines 241-243: Why were “lin” and “quad” described in parentheses, if “linearly” and “quadratic” were described in the description? The same for all the results description.

Answer: This has been revised.

Table 1 and 2: This table can be placed in the materials and methods topic.

Answer: Thank you for your suggestions. This has been revised. Please refer to line 138 and line 171.

For linear and quadratic effects, regression equations should be used to describe the results.

Tables are not complete; these do not show p-values for comparisons of means or p-values for regression. Tables should present both p-value information for comparison of means and for regression analysis.

Answer: Thank you for your suggestions. This has been revised accordingly.

Discussion: This discussion follows a correct description of the discussion, theories, hypotheses, or claims about how the authors obtained their results. Furthermore, the authors should decrease comparison with many authors, mainly those who have different results due to the composition of the FAs.

Lines 315-316: Ok but the milk production was not affected.

Answer: Thank you for your suggestions. This has been revised accordingly.

Lines 321-326: That's true, but it needs to be described in detail. Also, given the other results obtained, are you sure that these three hypotheses are valid?

Answer: Thank you for your suggestions. This has been revised. Please refer to lines 364–368.

Lines 335-336: This is true; in this sense, why is the above description about lower milk production necessary?

Answer: Thank you for your suggestions. This sentence has been revised. Please refer to line 377.

Conclusion: To improve the conclusion, add feed efficiency or feed conversion ratio to the results to show that HSO was efficient.

Answer: Thank you for your suggestions. This has been revised. Please refer to line 536.

References: More than 50% of references are older than 10 years. Update references to reduce this value to around 20%.

Answer: Thanks you for your suggestions. This has been revised. Please refer to the References section of the manuscript.

Reviewer 2 Report

Comments and Suggestions for Authors

MS ID: animals-3414956

Brief Abstract: In the article 'Effect of hemp seed oil on milk performance, blood parameters, milk fatty acid profile, and rumen microbial populations in dairy buffaloes: A preliminary study', the authors analyse the effects of dietary hemp seed oil (HSO) on dairy buffaloes, hypothesising that dietary HSO may improve milk performance, alter antioxidant and blood lipid levels, modify rumen bacterial structure, and increase the percentage of milk functional fatty acids (such as saturated and unsaturated C18 fatty acids) in milk from dairy buffaloes. The results show an improvement in the antioxidant capacity of whey and regulation of lipid metabolism following dietary supplementation with HSO. In particular, the authors found that HSO supplementation increased the percentage of functional milk fatty acids (e.g. omega-6 and omega-3 PUFA) in individual milk samples. The results of the study provide a basis for a natural additive to improve milk quality and health in dairy buffaloes.

I found the paper interesting and in line with the aims of the journal, but I have a number of suggestions for the authors, which I will list below, point by point:

Lines 28-30: To achieve this aim, we used seventeen healthy four-year-old dairy buffaloes divided into three groups and fed the following diets: (1) no HSO supplement (H0, n = 6), (2) a supplement of 100 g/d HSO (H1, n = 5), and (3) a supplement of 200 g/d HSO (H2, n = 6). The total duration of the experiment was 42 days, including a 15-day pre-feeding period.

Seventeen dairy buffaloes (indicate buffalo breed) were divided into 3 groups (indicate number of animals per group), see also lines 123-126.

Lines 82-83: Hydrogenation by rumen bacteria is a pathway that affects the composition of milk fatty acids [14,15].

I suggest the authors explain the concept in more detail.

Lines 89-90: HSO and dairy buffalo are two specialties from Guangxi, China. 

This sentence is redundant and lacks a reference.

Lines 166-167: Before the morning feeding on day 41 of total experimental period, blood samples (10 mL) were collected from the carotid artery of the dairy buffalo.

I assume that the blood samples were collected by a veterinarian, please specify.

Lines 218-230: 2.7. Statistical analysis.

When groups of experimental animals are unbalanced (6, 5, 6), it would be appropriate to use non-parametric tests, although in this case the authors correctly used a post hoc test.

The correct form is 'Tukey-Kramer post hoc'.

Also report the SEM, standard error of the mean.

Lines 261-263: It can be seen from Table 6 that the observed bacterial species, the Chao richness estimator, the ACE estimator, the Shannon index and the Simpson index are not significantly different between the groups (p > 0.05).

A short explanation of the quoted indices is necessary

Lines 320: Tucumã possibly Astrocaryum spp, specify.

Line 487: changes in milk performance. what is meant by “milk performance”? Rewrite the sentence.

Table S1. Effects of HSO supplementation on dry matter intake and average milk yield and composition in dairy water buffaloes.

Insert unit of measurement for each parameter.

Author Response

Brief Abstract: In the article 'Effect of hemp seed oil on milk performance, blood parameters, milk fatty acid profile, and rumen microbial populations in dairy buffaloes: A preliminary study', the authors analyse the effects of dietary hemp seed oil (HSO) on dairy buffaloes, hypothesising that dietary HSO may improve milk performance, alter antioxidant and blood lipid levels, modify rumen bacterial structure, and increase the percentage of milk functional fatty acids (such as saturated and unsaturated C18 fatty acids) in milk from dairy buffaloes. The results show an improvement in the antioxidant capacity of whey and regulation of lipid metabolism following dietary supplementation with HSO. In particular, the authors found that HSO supplementation increased the percentage of functional milk fatty acids (e.g. omega-6 and omega-3 PUFA) in individual milk samples. The results of the study provide a basis for a natural additive to improve milk quality and health in dairy buffaloes.

I found the paper interesting and in line with the aims of the journal, but I have a number of suggestions for the authors, which I will list below, point by point:

Lines 28-30: To achieve this aim, we used seventeen healthy four-year-old dairy buffaloes divided into three groups and fed the following diets: (1) no HSO supplement (H0, n = 6), (2) a supplement of 100 g/d HSO (H1, n = 5), and (3) a supplement of 200 g/d HSO (H2, n = 6). The total duration of the experiment was 42 days, including a 15-day pre-feeding period.

Seventeen dairy buffaloes (indicate buffalo breed) were divided into 3 groups (indicate number of animals per group), see also lines 123-126.

Answer: This has been revised. Please refer to lines 28–30.

Lines 82-83: Hydrogenation by rumen bacteria is a pathway that affects the composition of milk fatty acids [14,15].

I suggest the authors explain the concept in more detail.

Answer: This has been revised. Please refer to lines 101–103.

Lines 89-90: HSO and dairy buffalo are two specialties from Guangxi, China. 

This sentence is redundant and lacks a reference.

Answer: Thank you for your suggestion. Based on the advice of reviewer 1 and this comment, this sentence has been deleted.

Lines 166-167: Before the morning feeding on day 41 of total experimental period, blood samples (10 mL) were collected from the carotid artery of the dairy buffalo.

I assume that the blood samples were collected by a veterinarian, please specify.

Answer: This has been revised. Please refer to lines 217–218.

Lines 218-230: 2.7. Statistical analysis.

When groups of experimental animals are unbalanced (6, 5, 6), it would be appropriate to use non-parametric tests, although in this case the authors correctly used a post hoc test.

The correct form is 'Tukey-Kramer post hoc'.

Also report the SEM, standard error of the mean.

Answer: This has been revised. Please refer to line 264 and line 270.

Lines 261-263: It can be seen from Table 6 that the observed bacterial species, the Chao richness estimator, the ACE estimator, the Shannon index and the Simpson index are not significantly different between the groups (p > 0.05).

A short explanation of the quoted indices is necessary

Answer: This has been revised. Please refer to line 330.

Lines 320: Tucumã possibly Astrocaryum spp, specify.

Answer: Thank you for your suggestions. According to reviewer 1 (who suggested that we decrease the amount of comparisons with many authors), this sentence has been deleted.

Line 487: changes in milk performance. what is meant by “milk performance”? Rewrite the sentence.

Answer: This has been revised. Please refer to line 527.

Table S1. Effects of HSO supplementation on dry matter intake and average milk yield and composition in dairy water buffaloes.

Insert unit of measurement for each parameter.

Answer: Thank you for your suggestions. This has been revised. Please refer to Table S1 in the Supplementary Materials (Table S1).

Round 2

Reviewer 1 Report

Comments and Suggestions for Authors

Dear authors, reviewers make suggestions with the aim of improving the manuscript; however, it is the decision of the authors to accept them in full, accept them partially or reject them. I am pleased with the responses; and the addition of the power test values ​​greatly improved the manuscript because it is now possible to identify the parameters that were definitely affected by the treatment.